# Mirror, Mirror on the Wall: Automating Dental Smile Analysis in Smart Mirrors with CNN and Diffusion Model

**Mariia Baidachna** [* 1]   **Haneen Fatima** [2]   **Rahaf Omran** [3]   **Muhammad Ali Imran** [2]   **Ahmad Taha** [2]
**Lina Mohjazi** [* 4]

## Abstract

This paper presents a smart diagnostic framework for dental smile analysis. To accurately and efficiently identify esthetic issues from a single image of a smile, a convolutional neural network (CNN) was trained. To overcome the limitations of scarce data, a diffusion model was employed to generate dental smile images in addition to manually curated data. The CNN was trained and evaluated on three datasets: all real images, all generated images, and a hybrid dataset of equal proportions of real-to-generated images. All three models demonstrate accuracy significantly above the baseline in detecting excessive gingival display, unlocking a novel diagnostic method in smile analysis. Notably, the hybrid model achieved the highest accuracy of 81.61% ($p$ value $< 0.01$), highlighting the effectiveness of generative data augmentation for machine learning. The proposed solution could be part of a standalone home-deployed smart mirror or connected to a network of an innovative Internet-of-Mirrors to facilitate patient-dentist communication.

## 1. Introduction

The rapid growth of connected devices manifested by the Internet-of-Things (IoT) and the expanding capabilities of artificial intelligence (AI) have set the stage for an accelerated growth of digital beauty and healthcare (Senbekov et al., 2020; Al-Quraan et al., 2022). However, a significant gap is evident between the technological advancements of IoT and machine learning (ML) and the practical applications within the dental sector, especially esthetic smile analysis. There is

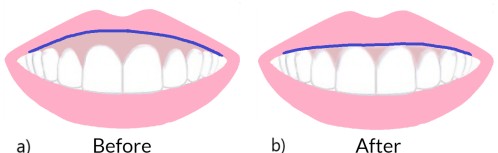

*Figure 1.* Presence of excessive gingival display in (a) before orthodontic treatment. An ideal lip line after treatment shown in (b). The blue line shows the location of the lip line.

an escalating demand for smile rectification (Armalaite et al., 2018) due to the impact a person's smile has on periodontal health (Maria Luca & Calandra, 2014), attractiveness, self-confidence, success and intelligence perception, and psychological well-being (Armalaite et al., 2018; Lukez et al., 2015). Orthodontic treatment of smile esthetic issues results in an improvement of many of these variables (May et al., 2019; Saini et al., 2022). However, before a patient can undergo treatment, the specific elements of the smile that require correction must be identified.

At present, the evaluation of esthetic appeal is confined to human eye assessment of smile elements (Sajjadi et al., 2016). One such element, the lip line, pertains to the extent of maxillary teeth displayed upon smiling (Khan et al., 2020). An ideal lip line exposes a range of three-quarters to the full clinical crown of maxillary teeth and only the interdental gingiva, while a high lip line reveals a continuous band of gingiva above the full clinical crown (Desai et al., 2009). Figure 1 shows a high lip line before and after treatment. Among esthetic issues, excessive gingival display was selected as the focus of this study due to its widespread occurrence, significant influence on smile attractiveness, and recognition as the most distracting element by dental students (Akyalcin et al., 2014; Ríos, 2020; Armalaite et al., 2018).

### 1.1. Drawbacks of Traditional Smile Analysis

Biases such as subjective esthetic appreciation and the speciality of the assessing dentist can affect smile assessment (Sajjadi et al., 2016). Different aspects of the smile are noticed by dentists of different specialties (Sajjadi et al., 2015).

[1]School of Computing Science, University of Glasgow, Glasgow, United Kingdom [2]James Watt School of Engineering, University of Glasgow, Glasgow, United Kingdom [3]Clinical Specialist at Align Technology, Solihull, United Kingdom [4]**AUTHORERR: Missing \icmlaffiliation.** . Correspondence to: Mariia Baidachna <baidachnasm@gmail.com>.

*Accepted at the 1st Machine Learning for Life and Material Sciences Workshop at ICML 2024. Copyright 2024 by the author(s).*

This discrepancy in clinical observations poses the question of whether general dentists or newly graduated clinicians without a substantial experience in practice might benefit from having "another set of eyes" when making complex clinical decisions.

Another drawback of the conventional methods is the delay between patients initiating appointments and receiving rectification, sometimes taking months before redirection to a professional (Prasanna & Vignesh, 2019). Long waiting time is a major factor that negatively affects patient experience and patient-provider relationship (Inglehart et al., 2016). Regular review appointments are vital for monitoring treatment progress, suggesting the need for a streamlined system to connect patients with healthcare providers and reduce waiting times. Similar challenges arise in the broader medical sector due to the workforce deficit in the UK National Health Service, which has been struggling to recover post the COVID-19 pandemic, resulting in the worst-recorded waiting times for cancer care (Aggarwal et al., 2024). This trend is reflected globally with the shortage of health care workers in future years projected to be higher in post-pandemic calculations (Downey et al., 2023). To counter this, there has been a drastic shift towards remote consultation and digital healthcare (Susilo et al., 2021; Shilpa & Kaur, 2022). However, smile analysis is lagging behind. To the best of our knowledge, there are no reliable, non-invasive, time-efficient diagnostic tools available for smile esthetics evaluation.

### 1.2. Contribution and Vision

The limitations of existing dental esthetic diagnosis methods prompt consideration of novel solutions. For instance, ML is currently employed in numerous healthcare fields and is seen to possess the ability to improve accuracy, increase efficiency, and aid in decision-making processes, leading therefore, to a possible revolutionization of daily practices and positive societal and environmental impacts (Dhopte & Bagde, 2023). Moreover, wireless communication networks pave the way for enhanced connectivity, enabling a multitude of applications that can revolutionize the field of diagnostic tools by ensuring efficiency and timely delivery of results through the IoT (Al-Quraan et al., 2022). Various areas have benefited from connected technology, where IoT systems serve as the backbone of smart devices, and integrating the IoT into healthcare has the power to improve the quality of life (Mohanty et al., 2016; Aghdam et al., 2021). We hypothesize that combining the IoT with ML and big data analysis can non-invasively resolve many limitations of traditional healthcare, particularly within smile analysis.

In this paper, we propose a novel diagnostic tool for excessive gingiva detection, leveraging a sequential topology CNN model. A persistent challenge in our research stems

from the insufficient number of data available to generalize to unseen content. To resolve this issue, we curate a dataset of a total of 512 dental smile images from available images in the open source and text-to-image AI-generated samples. The inclusion of diffusion model (DM) generated data significantly enhances classification accuracy, demonstrating the efficacy of our methods. The usage of text-to-image generative models in data augmentation is still an early practice, but we show that it has potential in dentistry. It enables us to attain higher quality results, for example, detecting excessive gingival display correctly on previously unseen images with 81.61% accuracy, above the 50% baseline accuracy.

Furthermore, we integrated the trained CNN model into a minimalistic user interface (UI) that guides the user to the correct position and captures their smile to ease the transition between theoretical and practical applications. The image is subsequently preprocessed according to specifications and passed through the pre-trained CNN model. The image is then analyzed in the backend, and the UI displays the result of the smile analysis obtained from the model in real-time, forming an end-to-end application.

This application could be stand alone or integrated into an innovative IoT smart mirror technology, termed as the Internet-of-Mirrors (IoM) (Fatima et al., 2023). This visionary ecosystem of interconnected smart mirrors could integrate the smile analysis application, as well as other digital health and beauty applications. The motivation lies in connecting patients with healthcare professionals or products in a timely manner, as depicted in Figure 2. In this system, each mirror would be equipped with sensors, such as a camera, to acquire user input. In the context of smile analysis, the mirror would take the dental smile image, crop it according to the specifications, and use pre-trained CNN model proposed in this paper to produce an output and inform the patient of their smile condition. The results of potential dental issues and personalized suggestions would be displayed on the dashboard. Smart mirrors placed in homes would serve users by promptly delivering tailored products or healthcare recommendations to align with their individual needs, and effectively facilitating connections between patients and professionals. Furthermore, smart mirrors placed in clinics could resolve the esthetic subjectivity issue by providing a basis for the diagnosis.

To summarize, the main contributions of the paper are the following:

- **Novel Diagnostic Smile Analysis Tool:** A CNN model trained on real and AI generated images of dental smiles to detect excessive gingival display, providing an non-invasive digital diagnostic tool.

- **Dental Smile Dataset:** A dataset of 512 labeled dental smile images with a mix of real and AI-generated

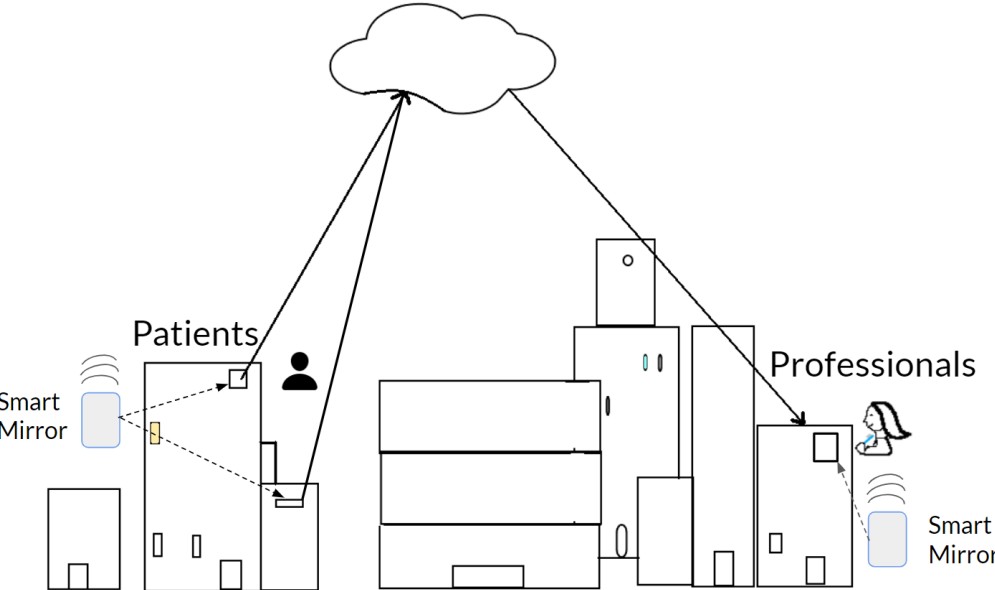

*Figure 2.* Interaction possibilities between professional dentists and patients as proposed in (Fatima et al., 2023). The diagram depicts each patient equipped with a smart mirror that delivers information to dentists over a wireless network. The patient can then be redirected to a professional through booking an appointment. The professionals can also rely on smart mirrors for providing a basis in subjective diagnosis. This vision of interconnected dentistry would unlock the potential of building smart cities through a similar system.

images from a DM.

- **IoM Integration Vision:** Proposed solution of integration into an visionary IoM system, offering a smooth patient-provider interaction experience and instantaneously displaying the results of the diagnosis with treatment options.

### 1.3. Related Work

Previous studies that employed DL in dental analysis have adapted variable architectures with varying degrees of success. The study in (Zhu et al., 2020) chose a Mask Region-based CNN (R-CNN). Due to the absence of publicly available datasets, (Zhu et al., 2020) collected and annotated only 100 images, achieving a pixel accuracy of tooth segmentation between 90.1% and 97.4% for natural teeth and lower for dentures. The observed high accuracy is consistent with the nature of Mask R-CNN methods, which prioritize performance over speed (Soares et al., 2019). In pursuit of scalable real-time results and a smooth user experience, the study conducted by (Lee & Kim, 2022) employed an extension of You Only Look Once (YOLO), the YOLACT++ instance segmentation model, analyzing a dataset comprising 5500 images of faces distributed across diverse classes. While the model exhibited high accuracy in detecting facial features like the nose, eyebrows, and eyes, a significant decline in precision was observed when segmenting the gingiva and buccal corridor. This decrease in precision may be attributed to a relatively lower number of annotated instances per class

for these specific regions in the training dataset, highlighting the importance of a well-balanced dataset.

The common thread in these studies is the use of relatively small datasets, which leads to poor generalization (Motamed et al., 2021). To counter this, numerous studies have been taking advantage of generative AI (GenAI) for image generation, such as generative adversarial networks (GANs), to augment small or imbalanced datasets. The study in (Alauthman et al., 2023) used 13 small medical datasets to show that the optimal proportion of GAN augmentation significantly enhances performance of all ML classifiers. Similarly, (Zhu et al., 2017) built an emotion classification model and used GAN augmentation to increase samples of less common classes such as disgust, increasing classification accuracy by 5% - 10%. The study in (Motamed et al., 2021) observed a similar trend, employing GAN data augmentation to amplify an X-ray dataset and improving the CNN model for pneumonia and COVID-19 detection.

In recent years, DMs have been shown to generate superior images to GANs, becoming the state-of-the-art generative models (Dhariwal & Nichol, 2021). The paper in (Zhang et al., 2023) experimented with GANs and DMs for pneumonia ultrasound image augmentation and observed an improvement in both detection accuracy and generative image quality with DM-based augmentation.

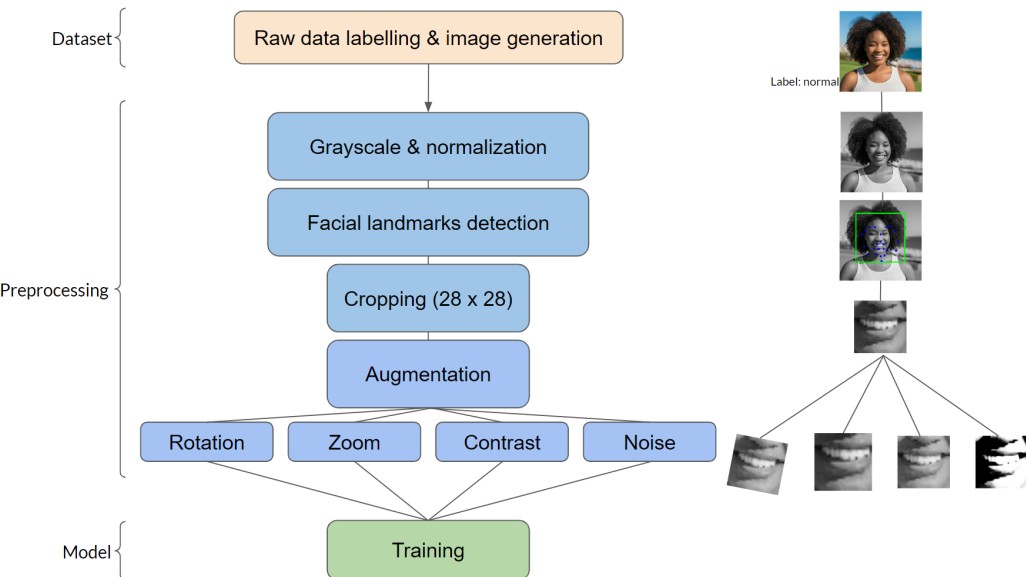

*Figure 3.* Diagram of the pipeline from data collection to preprocessing to CNN training. On the right is a visualization of the process a single image goes through. The image used is a random sample image from the Adobe Firefly AI-generated dataset with some of the included data augmentations.

## 2. Methodology

### 2.1. Dataset

In the context of dental smiles, the absence of a suitable dataset in open-source repositories and the ethical collection of a diverse array of dental images encompassing proportional ethnicities, genders, and degrees of gingival display became a persistent endeavor posed a significant challenge.

In total, two sets of 256 (counting 512 combined) images were collected and evenly distributed into "gummy" and "normal" classes. The first set was manually curated from publicly available pictures scrapped from online frontal face images of people smiling. These included highly positive excessive gingival display samples as well as more discrete ones. The second set was collected from text-to-image AI generated images using Adobe Firefly's DM (ado). The text prompts for the "normal" class varied from "frontal portrait of a person smiling with teeth; this person does not have a gummy smile or braces" to simply picking out relevant images in "frontal portrait of a person smiling". Despite some filtering being required, this process still consumed less time and resources than manual annotation.

Both sets contributed 56 randomly chosen images each to the testing dataset, setting aside 112 of the original 512 images for testing. The remaining 400 (200 from real, 200 AI-generated by the DM) images were randomly split with 80% assigned to training and 20% to validation. In total, three separate datasets were curated: real images (256 images), AI-generated images (256 images), and combined images (512 images). The data was balanced evenly between the two classes, and the correctness of the labels was validated by a professional dentist.

### 2.2. Preprocessing

To mitigate overfitting, all images were normalized and grayscaled. Subsequently, each image underwent a process where a pre-trained facial landmark detector from the Dlib library was utilized to pinpoint the coordinates of the mouth, followed by cropping every image to a 28 by 28 pixels centered around the mouth. This systematic preprocessing step that remained consistent on training, validation, and testing data contributed to the overall quality and consistency of the dataset.

The pipeline of the entire process is presented in Figure 3 as a flowchart from receiving raw images to preprocessing to training with an example of a "normal" smile generated Adobe Firefly image.

### 2.3. Model Architecture

A CNN classifier model was built using Tensorflow and keras to classify previously unseen images into either "gummy" or "normal" smile classes. A sequential model architecture was selected for its simplicity, offering a linear topology that enables the customization of layer stacking while maintaining one input tensor and one output tensor. This choice is driven by the primary goal of minimizing the risk of overtraining, as small datasets have a tendency to not

generalize to unseen testing data. The overall architecture of the model follows a standard layer sequence. The pattern of 2D convolution layer followed by 2D max pooling is used twice. The rectified linear unit (ReLU) function, represented by $R(z) = \max(0, z)$, served as the activation parameter for the first layer, and a softmax function was used to output the probability distribution for the two classes.

Each model, distinguished by real, AI-generated, and combined datasets, was trained using the Adam optimizer. Through hyperparameter tuning, the optimal number of epochs was determined to be at 150 and a batch size 32. All parameters remained constant throughout training sessions.

## 3. Results

The research yielded several key findings. First, we evaluate the performance of the conventional CNN model for detection of excessive gingival display. The model employing DM data augmentation in addition to geometric transformations achieved an average accuracy of 81.61% with a standard deviation of 1.94. This statistic reflects the average accuracy across 10 trials, each involving training the model for 150 epochs and evaluating its performance on previously unseen images to ascertain the accuracy of label predictions.

The CNN model that was trained on images generated from the DM was compared to the two byproduct models of 200 real images and 200 AI-generated images. For consistency, each model's testing accuracy was averaged from 10 trials. The box plots of each model are shown in Figure 4. As can be seen from the error bars on the graph and the $p$ value greater than 0.1, there was no significant difference between the performance of the model trained on real human images and the model trained solely on generated images. However, the median accuracy of the model trained on real and generated data was significantly higher than both the real and GenAI images separately ($p$ value $< 0.01$) and the baseline of 50% ($p$ value $<< 0.01$). These results suggest that incorporating DM-generated data, which is less costly to collect than real data, substantially enhances classification accuracy.

Additionally, the confusion matrix of a representative model with an accuracy of 82.14% and an F1 score of 0.82, indicating well-balanced performance. Figure 5 shows that the false positives and false negatives on the upward diagonal are generally outnumbered the correctly labeled testing data points. This shows the importance of curating a balanced dataset, a task made more feasible with the use of DM-generated data for the more rare "gummy" class.

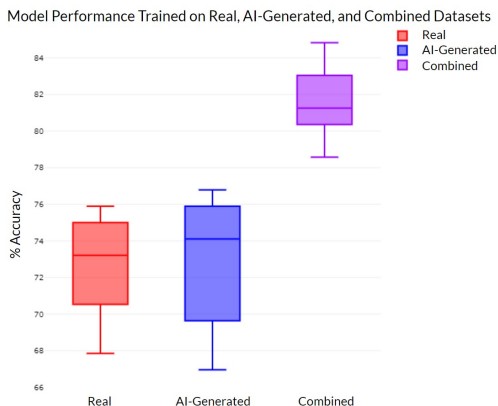

*Figure 4.* Box plots of the testing accuracy of models trained on real, AI-generated, and combined datasets.

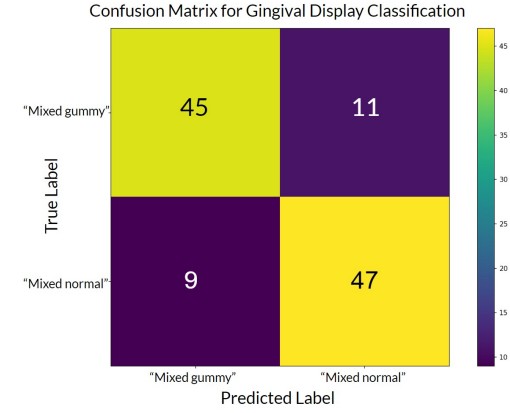

*Figure 5.* The confusion matrix of the CNN model results.

## 4. Discussion and Conclusion

We opted to evaluate each model using a combined dataset comprising both real and AI-generated images, rather than relying solely on real images. This decision stems from the recognition that both types of images may harbor biases, though potentially in divergent directions. The manually curated 'real' dataset exhibits significant variation in image quality but may lack proportional ethnic representation, potentially resulting in the model's underperformance for minority groups. Conversely, the DM yielded diverse images, but it may lack certain nuanced human characteristics, the absence of which could disrupt the model's accuracy. Lacking a precise quantitative method to gauge these biases, it would be imprudent to assume that either dataset thoroughly captures reality. While the ideal future testing data would mirror reality in terms of camera quality, lighting, and angle, our current approach involves making estimations based on the combined dataset, relying on data augmentation to compensate for potential inconsistencies.

We have not only presented a novel framework that resolves all the current drawbacks of traditional smile analysis, but also provided a high-performing model and a dental smile dataset. The research represents a significant step forward in applying ML to dental smile analysis, underscoring the potential of utilizing IoT and ML techniques in real-time healthcare applications.

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
