# OpenReview forum: "Mirror, Mirror on the Wall: Automating Dental Smile Analysis in Smart Mirrors with CNN and Diffusion Model"
_ICML.cc/2024/Workshop/ML4LMS — ML4LMS Poster_

### Official Review · Reviewer_ee5w · 2024-06-01
**Automating Dental Smile Analysis with CNN and Diffusion Model in Smart Mirrors**

**Rating:** 8
**Confidence:** 4

**Review:**

The paper presents an innovative approach to automating dental smile analysis using a convolutional neural network (CNN) and a diffusion model for data augmentation. The proposed method addresses the subjectivity and latency of traditional smile assessments by leveraging AI-generated images to enhance classification accuracy. The CNN trained on real and AI-generated data achieved 81.61% accuracy in detecting excessive gingival display. This solution has potential applications in standalone smart mirrors or an Internet-of-Mirrors network, facilitating patient-dentist communication.

Review:

Pros:

Novel application of CNN and diffusion models for dental analysis. Effective use of AI-generated data to improve model performance.
Potential for real-time, non-invasive dental diagnostics.

Cons:

Limited dataset size may affect generalization. Needs further validation in diverse real-world scenarios.

---

### Official Review · Reviewer_SrEn · 2024-06-12
**Authors Fail to Support Main Claim**

**Rating:** 4
**Confidence:** 1

**Review:**

I am not able to judge the originality and significance of this work in the field of dental smile analysis which is why this review focusses on the papers clarity and quality.

Abstract:
- "This paper presents a framework for a smart diagnostic tool for dental smile analysis" doesn't give a good easily understandable summary of the approach presented.
- "impressive accuracy of 81.61% in detecting excessive gingiva display": impressive is an opinion and thus subjective.

Contribution and Vision
- "Proposed solution of integration into an visionary IoM system, offering a smooth patientprovider interaction experience and instantaneously displaying the results of the diagnosis with treatment options.": This reads like an advert and should be considered to be rewritten.

Content
-The main claim of the paper is that dataset augementation via Text-To-Image Generated Images improves accuracy on real-world data yet they do not report the performance on the real world images alone but only on a mixed set of images containing AI-generated as well as real-world images. This means they do not provide clear evidence for their main claim which makes it hard to argue for an acceptance.

---

### Official Review · Reviewer_PpGi · 2024-06-12
**Nicely written paper with room for extension**

**Rating:** 8
**Confidence:** 4

**Review:**

This paper is about a CNN model to classify "smiles" into "normal smiles" vs "gummy smiles". This might be useful for dental applications.

I doubt this can actually help dentists, nonetheless it can help initial screening of patients or self-diagnositic. However, the point of this paper is to use diffusion models to generate 256 frontal face portraits, together with 256 collected real pictures to form a overall dataset for the training of CNN model. The paper compared the cases of building a CNN model in linear topology, but trained using (a) generated images only (b) real images only (c) a mix of both real and generated images. The models are trained for 150 epochs. And it is found that the case (c) of mixed training data performs significantly better.

The merit of this paper is as a prove that the modern invention of diffusion models can help the aged CNN models by providing dataset for training at low cost. However, I would question, or the authors have not clarify, that the better result of "mixed model" is due to a larger dataset, because I note that the author mentioned 56 pictures are contributed to training from both sets. Does it means a mixed model is using 112 pictures while the other two models are half of this size? Training time measured in number of epochs is hiding this difference. A better way is to measure how many steps you trained it.

As an extension, and make the contribution generalizable to other applications, it would be nice to investigate in the following direction:

- does a mixed dataset help converge faster? e.g., for the same validation set, you saved X% training steps to achieve the benchmark accuracy of Y
- what is the reason for mixed dataset helps? Is it working like an addition augmentation dimension?
- does it work for different network topology/architecture? e.g., how can this result generalized for deeper/shallower network? How about pure-CNN network vs CNN+dense layers?